# Fabrication of Zn$^{2+}$-Loaded Polydopamine Coatings on Magnesium Alloy Surfaces to Enhance Corrosion Resistance and Biocompatibility

Lingjie Meng [1,†], Xuhui Liu [2,†], Qingxiang Hong [1], Yan Ji [1], Lingtao Wang [1], Qiuyang Zhang [1], Jie Chen [1,*] and Changjiang Pan [1,*]

1 Faculty of Mechanical and Material Engineering, Jiangsu Provincial Engineering Research Center for Biomaterials and Advanced Medical Devices, Huaiyin Institute of Technology, Huai'an 223003, China; mlj963555816@163.com (L.M.); 15715183778@163.com (Q.H.); jy18362965023@163.com (Y.J.); wjjswlt@126.com (L.W.); qyzhang@hyit.edu.cn (Q.Z.)
2 The Affiliated Huai'an Hospital, Xuzhou Medical University, Huai'an 223003, China; liuxuhui_0221@163.com
* Correspondence: jiechen@hyit.edu.cn (J.C.); panchangjiang@hyit.edu.cn (C.P.)
† These authors contributed equally to this work.

**Abstract:** In this study, inspired by the adhesion protein of mussels, a Zn$^{2+}$-loaded polydopamine (PDA/Zn$^{2+}$) coating was prepared on an alkali–heat-treated magnesium alloy surface, through the chelating effect of PDA with metal ions, to improve anticorrosion and biocompatibility. The results of water contact angles show that the PDA/Zn$^{2+}$ coatings with different Zn$^{2+}$ contents had excellent wettability, which contributed to the selective promotion of the albumin adsorption. The corrosion degradation behaviors of the modified magnesium alloys were characterized using potentiodynamic scanning polarization curves, electrochemical impedance spectroscopy (EIS), and an immersion test, the results indicate that anticorrosion was significantly improved with the increase of Zn$^{2+}$ content in the coating. Meanwhile, the PDA/Zn$^{2+}$ coatings with different Zn$^{2+}$ concentrations demonstrated improved hemocompatibility, confirmed by assays of the hemolysis rate and platelet adhesion behaviors. In addition, the results regarding the growth behaviors of endothelial cells (ECs) suggest that, due to the sustained release of Zn$^{2+}$ from the coatings, the modified magnesium alloys could enhance the adhesion, proliferation, and upregulated expression of vascular endothelial growth factor (VEGF) and nitric oxide (NO) in endothelial cells, and that better cytocompatibility to ECs could be achieved as the Zn$^{2+}$ concentration increased. Therefore, the PDA/Zn$^{2+}$ coatings developed in this study could be utilized to modify magnesium alloy surfaces, to simultaneously impart better anticorrosion, hemocompatibility, and endothelialization.

**Keywords:** magnesium alloy; zinc ions; corrosion resistance; endothelialization; biocompatibility

## 1. Introduction

Percutaneous coronary intervention (PCI) with stent implantation is an extensive and effective method for treating stenotic cardiovascular disease; however, the biocompatibility of traditional metal stents within current clinical practice is limited. The drugs released from the drug-eluting stents (DESs) can prevent excessive neointimal hyperplasia, to reduce the in-stent restenosis; however, it can also inhibit endothelial repair and regeneration, leading to clinical complications such as chronic inflammation, late thrombosis, and late in-stent restenosis [1,2]. The drawbacks of current vascular stents have pushed researchers to explore novel stents made from biodegradable materials. Magnesium (Mg) is the fourth trace element in the human body, and plays important roles in impeding abnormal nerve excitation, participating in protein synthesis, abating hypertension, treating acute MI (myocardial infarction), and preventing atherosclerosis [3]. Cardiovascular stents made from a magnesium alloy have better biodegradation and mechanical properties. These

offer potential advantages in overcoming clinic complications caused by non-degradable metal stents, such as chronic inflammatory responses, late thrombosis, and long-term use of anticoagulants. In the past several years, many types of magnesium alloy with excellent mechanical properties have been developed. For example, Zhang et al. developed a Mg alloy with ultrahigh ductility (nearly 50%) by combining Er micro-alloying (0.3 at% Er) with the appropriate grain refinement (~8 μm) [4]. Zhao et al. [5] utilized super-defined wave-welding technology to achieve high-strength connections between two heterogeneous light alloys (magnesium alloy and copper alloy). However, most of these were quickly corroded in the complicated in vivo physiological environment, easily leading to the premature loss of mechanical support after implantation. Moreover, the limited bioactivities on the magnesium-based biomaterials' surfaces could not effectively regulate the physiological reactions of the ambient microenvironment after implantation, which may result in late endothelial healing and thrombosis. Therefore, when exploring new ways of maintaining good biocompatibility, concurrently controlling the degradation speed of magnesium-based alloys remains the primary focus, and represents the sticking-point in clinical practice regarding vascular stents made from magnesium and its alloys.

Given that corrosion resistance and biocompatibility are closely related to surface characteristics, surface modification is one of the operationally easy and effective approaches toward overcoming the clinical complications of magnesium alloys in cardiovascular stents. Currently, various methods have been explored to improve the anticorrosion and biocompatibility of magnesium alloys, including chemical and electrochemical treatments, the preparation of inorganic coatings (sol–gel coating, inorganic non-metallic coating, layered double hydroxide coating, etc.), the in-situ introduction of bioactive factors onto the surface, the construction of polymer coatings with or without biomolecules, etc. [6–12]. Although these methods can obviously reduce the biodegradation speed of magnesium and its alloys, and introduce chemical groups or bioactive factors onto the surface, as well as reducing the side effects after implantation to some degree, there are still shortcomings, including the complex preparation steps, the long time-consuming procedures, the short duration of action, the difficult performance control, and other issues.

It is well known that zinc is the second trace element in the human body, and has a significant impact on the biological behaviors of cells and tissues [13,14]. Zinc ions play a crucial role in preventing and treating various physiological diseases, such as preventing local ischemia and vascular infarction. Zinc ions have a cytoprotective effect that can accelerate the integrity of endothelial cells, thereby preventing atherosclerosis, and facilitating the regeneration of the damaged vascular endothelium [15], which plays an important role in maintaining the normal physiological function of the human vascular endothelium. In this study, polydopamine coatings containing different $Zn^{2+}$ concentrations (PDA/$Zn^{2+}$) were prepared by a one-step process on the alkali–heat-treated magnesium alloy surface, and electrochemical degradation behaviors, blood compatibility, and endothelial cell growth behaviors were investigated in detail. The results indicate that the introduction of $Zn^{2+}$ onto the magnesium alloy's surface can effectively enhance anticorrosion and anticoagulation, and promote endothelial cell growth for endothelialization.

## 2. Materials and Methods

### 2.1. Construction of PDA/$Zn^{2+}$ Coatings

AZ31B magnesium alloy plates (12 mm diameter, 4 mm thickness) were polished to the point of displaying no scratches. The polished plates were ultrasonically cleaned for 10 min by acetone and ethanol, respectively. The cleaned Mg plates were immersed in a 3 M NaOH solution to be treated for 24 h at 75 °C. After being washed and dried, the alkali–heat-treated samples (Mg-OH) were dipped into the mixed solutions of 2 mg/mL dopamine and 1 mg/mL, 2 mg/mL, and 3 mg/mL ZnSO$_4$.7H$_2$O (Tris-HCl buffer, pH 8.5, volume ratio 1:1), respectively, for 4 h at room temperature. The as-prepared samples were named as Mg-PDA/1Zn, Mg-PDA/2Zn, and Mg-PDA/3Zn, respectively.

## 2.2. Surface Characterization

The surface chemical structures of the different samples were characterized by attenuated total reflection–Fourier-transform infrared spectroscopy (ATR-FTIR, TENSOR27, Bruker of Germany, Mannheim, Germany) and X-ray photoelectron spectroscopy (XPS, Quantum 2000; PHI Co., Chanhassen, MN, USA). The surface morphologies were examined using scanning electron microscopy (SEM, FEI, Quata 250, Portland, OR, USA). The surface wettability was characterized by the water contact angles of three parallel samples; the average value for each sample was calculated and expressed as mean ± standard derivation (SD).

## 2.3. In Vitro Electrochemical Corrosion Degradation Behavior

### 2.3.1. Potentiodynamic Scanning Polarization Curves and EIS

The three-electrode system (with the sample as the working electrode, platinum wire as the auxiliary electrode, and Ag/AgCl as the reference electrode) was utilized to measure the potentiodynamic scanning polarization curves and electrochemical impedance spectra (EIS) of the different modified samples. The test was carried out using a CHi660D electrochemical workstation (CHI Instruments, Inc., Shanghai, China). The test solution was Hank's simulated body fluid (SBF). Before the test, the sample was immersed into SBF to obtain the stable open circuit potential, and then the scanning polarization was carried out, at a scan speed of 1 mV/s, to obtain the potentiodynamic scan polarization curves. The corrosion current density and potential were obtained using the Tafel method, and the corrosion depth per year was determined using the Equation (1) [16]:

$$d = 3.28 \times 10^{-3} \, (M/n\rho)I_{corr} \tag{1}$$

where d represents the corrosion depth per year (mm/y); M and ρ are equal to the gram atomic weight (24 g/mol) and the density (1.74 g/cm$^3$) of Mg; n represents the number of electrons lost by Mg ($n = 2$); and $I_{corr}$ is the corrosion current density (μA/cm$^2$).

A sinusoidal alternating current with an amplitude of 10 mV was used to carry out the EIS measurement from high frequency to low frequency ($10^5$–0.1 Hz), and the impedance parameters of the equivalent circuit were fitted using the Zview software. The polarization resistance ($R_p$) was obtained from the EIS.

### 2.3.2. Immersion Experiment and pH Changes

The sealed sample with a 1 cm$^2$ exposed area was placed into 20 mL SBF solution (pH 7.4) for 1, 3, 7, and 14 days, respectively. The SBF solution was renewed every two days. The samples were rinsed using deionized water, and dried using compressed air flow. The corrosion morphologies and elemental compositions of the surface were examined by SEM and EDS (energy dispersive spectrometer), respectively. Meanwhile, the pH value for each sample solution was determined by a pH meter at predetermined times; three parallel specimens were measured, and the values were averaged.

## 2.4. Protein Adsorption

The bovine serum albumin (BSA) and fibrinogen (FIB) adsorption were evaluated using a BCA assay. The sealed magnesium alloy was first sterilized for 12 h under ultraviolet light, and then incubated for 2.5 h with 2 mL BSA and FIB solutions (1 mg/mL) at 37 °C, respectively. The sample was rinsed using phosphate buffer solution (PBS), followed by ultrasonically desorbing the adsorbed proteins for 30 min with 2 mL sodium dodecyl sulfate solution (1 wt% SDS). For each sample, 200 μL eluent was mixed with 100 μL BCA working solution (reagent A: reagent B = 50:1) in a 96-well plate, and absorbance at 562 nm was measured. The protein adsorption amount was deduced based on the standard curve.

### 2.5. Blood Compatibility

2.5.1. Hemolysis

Healthy human blood was centrifuged for 10 min at 1500 rpm to obtain red blood cells (RBCs). The 2% RBC suspension was prepared using physiological saline, and the 2 mL suspension was incubated for 3 h with the sample at 37 °C. For the negative control and the positive control, respectively, 2% RBCs suspensions diluted with normal saline, and distilled water, were used. A 1 mL amount of the culture solution was centrifuged for 5 min at 3000 rpm, and 200 µL supernatant was used to detect absorbance at 545 nm using a microplate reader (Bio-Tek Eons). Equation (2) was utilized to determine the hemolysis rate:

$$\text{Hemolysis (\%)} = (A - A_2)/(A_1 - A_2) \times 100\% \tag{2}$$

where A, $A_1$, and $A_2$ represent the absorbance of the sample, the positive control, and the negative control, respectively.

2.5.2. Platelet Adhesion

The anticoagulated human blood was centrifuged for 10 min at 1500 rpm/min to obtain platelet-rich plasma (PRP). A 200 µL amount of PRP was incubated for 2.5 h with the specimen at 37 °C. The specimens were washed thrice using PBS to remove the non-attached platelets, then the attached platelets were fixed for 3 h using 2.5% glutaraldehyde at 4 °C. The attached platelets on the surface were successively treated using 50%, 70%, 90%, and 100% ethanol solutions, for 15 min each. After being dried at room temperature, the sample surface was sprayed with a gold layer, and the platelets were observed using SEM. Five different SEM pictures (×3000) were randomly selected, and their total adhered platelets counted. This was expressed as the number of adhered platelets per unit area.

### 2.6. Growth Behaviors of Endothelial Cells

2.6.1. Cell Adhesion and Proliferation

The specimens were first placed into a 24-well culture plate and sterilized for 12 h using ultraviolet light. Into each well, 0.5 mL endothelial cell suspension ($5 \times 10^4$ cells/mL, ECV304, Cobioer, Nanjing, China), and 1.5 mL culture medium were added, to incubate with the sample for 6 h and 24 h at 37 °C with 5% $CO_2$, respectively. After being rinsed using the normal saline solution, the adhered cells were fixed using 2.5% glutaraldehyde at 4 °C for 3 h. The attached cells were treated using 100 µL rhodamine (10 µg/mL) for 20 min and 100 µL 4,6-diamidino-2-phenylindole (DAPI, 500 ng/mL) for 10 min, respectively. The cell images were recorded using a fluorescent microscope (Carl Zeiss A2 inverted).

The CCK-8 method was used to characterize cell proliferation. The cells were incubated with the different specimens as described above for 6 h and 24 h, respectively. Subsequently, the specimens were transferred to a new plate, and 0.5 mL of CCK-8 solution (10% in cell culture solution with 10% fetal bovine serum) was added into each well, to incubate for 3.5 h. Finally, 200 µL culture solution was used for measuring the absorbance at 450 nm, three parallel specimens were measured, and the values were averaged.

2.6.2. Endothelial Growth Factor (VEGF) Expression

Endothelial cells (ECs) were incubated with the specimen as described above. An enzyme-linked immunosorbent assay was carried out to evaluate the VEGF concentration expressed by ECs on the surface. After the cell culture, for each sample, 40 µL diluent and 10 µL cell culture medium were mixed in a 96-well plate. The culture plate was sealed and incubated for 1 h at 37 °C. After rinsing five times, 50 µL enzyme-labeled reagent was added, to incubate for another 15 min, followed by the successive adding of 50 µL chromogenic reagent A, and 50 µL chromogenic reagent B, into each well, to incubate for 15 min in the dark at 37 °C. Finally, 50 µL stop solution was added, and absorbance at 450 nm was measured. The VEGF concentration was determined based on the standard curve. The values of the three parallel specimens were averaged.

### 2.6.3. Endothelial Nitric Oxide (NO) Expression

The NO expressed by the ECs was determined by the nitrate reductase method. Firstly, endothelial cells were inoculated on each surface to culture for 6 h and 24 h as described above, respectively, and then 50 μL supernatant was moved into the 96-well plate, followed by the successive adding of 50 μL Grignard reagent I and 50 μL Grignard reagent II. The absorbance value at 540 nm was determined, and the NO concentration was deduced based on the standard curve.

## 3. Results and Discussion

### 3.1. Surface Chemical Structure and Morphologies

Figure 1A shows the ATR-FTIR spectra of the different specimens. Obviously, no obvious infrared absorption peaks on the Mg surface can be detected. A strong absorption peak at 3700 $cm^{-1}$ can be observed on Mg-OH, which belonged to the adsorption peak of -OH, indicating that NaOH treatment can produce a $Mg(OH)_2$ coating on the surface. The existence of the oxide layer can enhance anticorrosion to some degree. After constructing the PDA/$Zn^{2+}$ coatings, the absorption peaks at 1530 $cm^{-1}$ and 3200 $cm^{-1}$, belonging to the adsorption of the -NH and -$NH_2$ groups, respectively, can be detected, suggesting that the amino groups were introduced onto the surface. The adsorption peaks at 1450–1625 $cm^{-1}$ for the C=C and N-H peak of the benzene ring [17], and the peak at 2980 $cm^{-1}$ for the -$CH_2$ group, are clearly observed, demonstrating that the polydopamine coating was successfully constructed on the surface.

Figure 1B,C are the XPS survey spectra and high-resolution spectra, respectively, of the different specimens. Table 1 shows the surface atomic compositions of the different magnesium alloys. Since the unmodified magnesium alloys could be oxidized in the natural environment, and had some carbon contaminants [16], the peaks of $Mg_{1s}$ (1305.91 eV), $Mg_{2p}$ (346.59 eV), $C_{1s}$ (284.72 eV), and $O_{1s}$ (532.75 eV) could be detected on the Mg surface. For Mg-OH, the $Mg_{1s}$ peak was weakened, and its content dropped from 47.3% to 9.6%; the $O_{1s}$ and $C_{1s}$ peaks were enhanced and their contents increased to 58.56% and 31.77%, respectively. According to the high resolution spectra of $Mg_{1s}$, the fitting peaks at 1304.65 eV and 1307.44 eV can be attributed to $MgCO_3$ and $Mg(OH)_2$, respectively, indicating that alkali–heat treatment can produce a $Mg(OH)_2$ passivation layer on the surface. The formation of $MgCO_3$ may be due to the fact that a small amount of $CO_2$ in the air diffused into the inner layer, and reacted with the magnesium substrate. For the PDA/$Zn^{2+}$ modified sample, the occurrence of $Zn_{2p}$ (1022 eV) and $N_{1s}$ suggested that the $Zn^{2+}$-loaded polydopamine coating was successfully prepared on the surface. As the concentration of zinc ions in the solution increased, the Zn concentration in the coating augmented, while the Mg content decreased, which was beneficial to improving corrosion resistance. As dopamine concentration was constant (2 mg/mL) in the process of preparing the coatings, there was no significant change in $N_{1s}$ content. The $C_{1S}$ high-resolution spectrum of Mg-PDA/$Zn^{2+}$ can be fitted into two peaks; i.e., C-O at 285.14 eV, and C-C at 284.75 eV. The two peaks of $O_{1S}$ at 530.28 eV and 531.78 eV were mainly due to C=O and Zn·· O=C; the $N_{1S}$ high-resolution spectrum shows that the absorption peaks of C-$NH_2$ and $NH_3$ appeared at 400.41 eV and 398.75 eV, respectively [18]; these results indicated that the $Zn^{2+}$-loaded polydopamine coating had been successfully constructed on the surface.

The surface morphologies of the different specimens characterized by SEM are exhibited in Figure 2. The surfaces of the Mg and Mg-OH (data not shown for Mg-OH) were relatively smooth. For Mg-PDA/Zn, white particles can be seen on the surface, and the surface became a little rough; this is because the self-polymerization of dopamine under an alkaline environment can cause PDA aggregates to precipitate on the surface, resulting in an increased surface roughness [19]. Therefore, it can be seen from Figure 2 that particle agglomeration appeared on all the Mg-PDA/Zn surfaces. The dopamine concentration was constant; more dopamine would be used for chelating $Zn^{2+}$ with the increase in $Zn^{2+}$ in the solution, leading to the decreased white particles, which contributed to the production of a denser coating to enhance anticorrosion.

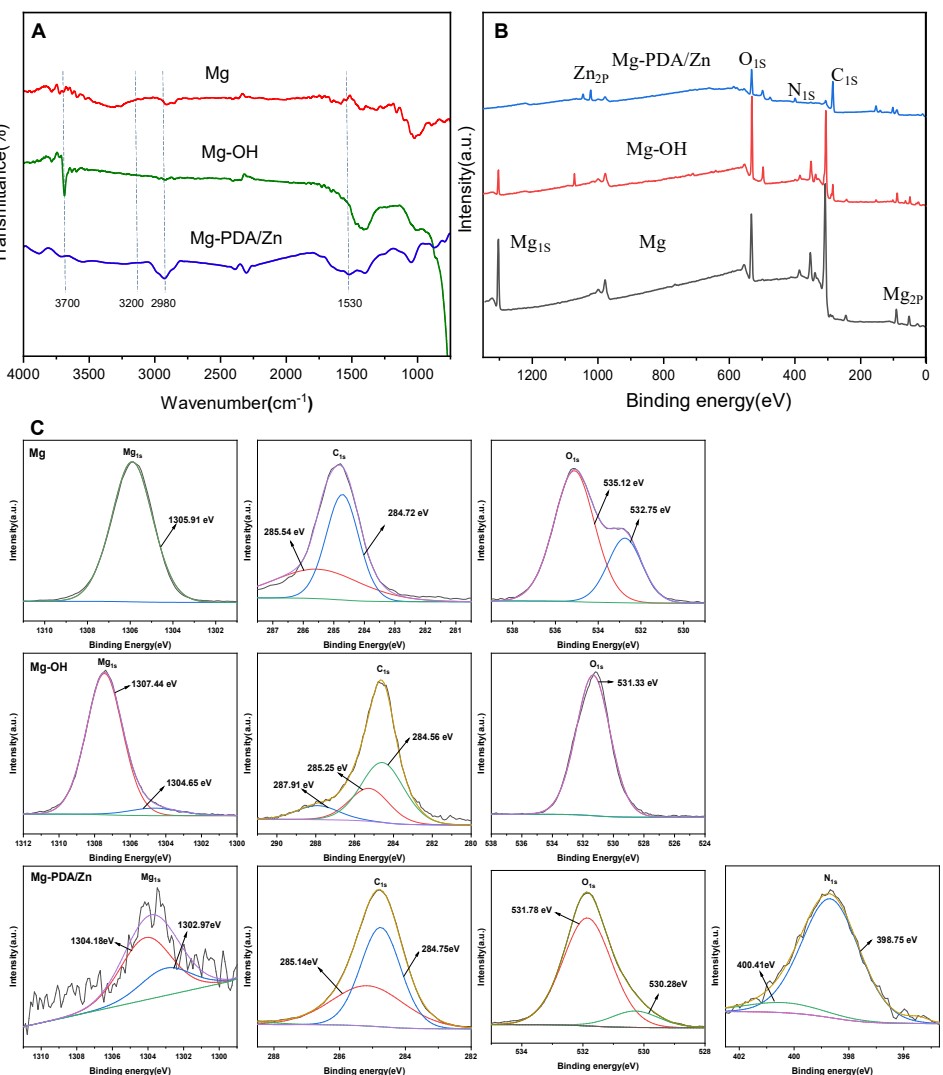

**Figure 1.** (**A**) ATR-FTIR, (**B**) XPS survey spectra, and (**C**) high-resolution XPS spectra of the different specimens.

**Table 1.** Atomic compositions of the different sample surfaces.

| Samples | Mg | C | O | Zn | N |
|---|---|---|---|---|---|
| Mg | 47.3 | 6.5 | 46.2 | - | - |
| Mg-OH | 9.6 | 31.8 | 58.6 | - | - |
| Mg-PDA/1Zn | 9.1 | 49.1 | 31.9 | 3.3 | 6.6 |
| Mg-PDA/2Zn | 7.3 | 53.3 | 28.5 | 5.2 | 5.7 |
| Mg-PDA/3Zn | 4.5 | 61.2 | 20.9 | 7.8 | 5.6 |

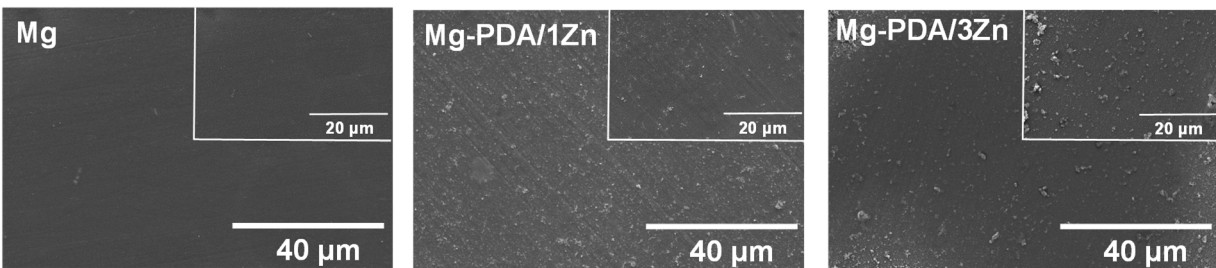

**Figure 2.** The representative SEM pictures of the different modified magnesium alloys.

### 3.2. Electrochemical Corrosion Degradation Behaviors

Potentiodynamic scanning polarization curves and EIS were used to investigate the electrochemical behaviors of the magnesium alloys, and the results are displayed in Figure 3A,B. The corresponding corrosion parameters of the different specimens, including the corrosion potential ($E_{corr}$), corrosion current density ($I_{corr}$), and annual corrosion depth of the different samples, are displayed in Table 2. In order to quantitatively describe the EIS graph response, equivalent circuit diagrams of the response were constructed, as shown in Figure 3B(b1,b2), respectively. Table 3 shows the EIS fitting parameters of the different magnesium alloys. Generally speaking, under the same corrosive environment, a larger $E_{corr}$ represents more thermodynamic stability, and a smaller $I_{corr}$ means a slower corrosion degradation rate [20]. The results shown in Figure 3A and Table 2 clearly indicate that the pristine magnesium alloy showed the largest corrosion current density ($1.42 \times 10^{-5}$ A·cm$^{-2}$) and corrosion depth per year ($2.80 \times 10^{-1}$ mm/y), as well as the lowest corrosion potential ($-1.55$V). Moreover, Figure 3B and Table 3 show that the Mg specimen had the smallest capacitive ring, and the lowest impedance value (5900 Ω·cm$^2$), suggesting that the blank Mg alloy had the worst anticorrosion properties. After alkali–heat treatment, the $E_{corr}$ was elevated to $-1.50$ V, and the $I_{corr}$ and annual corrosion depth were reduced to $1.47 \times 10^{-6}$ A·cm$^{-2}$, and $7.95 \times 10^{-2}$ mm/y, respectively, suggesting that alkali heat treatment could enhance anticorrosion properties to some degree. After the PDA/Zn$^{2+}$ coating was constructed, the $E_{corr}$ continued to shift positively, and the $I_{corr}$ decreased by an order of magnitude compared with Mg-OH, indicating that the PDA/Zn coating can further improve anticorrosion properties. It can be concluded that a dense and compact coating on the surface can effectively prevent the penetration of corrosive ions, providing a better corrosion resistance. The polarization resistances of Mg-PDA/1Zn, Mg-PDA/2Zn, and Mg-PDA/3Zn were 16,161 Ω·cm$^2$, 18,095 Ω·cm$^2$, and 57,422 Ω·cm$^2$, respectively, indicating that the increase in zinc ions in the coating was helpful for improving corrosion resistance. On the one hand, the zinc-loaded polydopamine coating could completely cover the magnesium alloy surface; on the other hand, the higher content of Zn$^{2+}$ could react with OH$^-$ to form a stable and dense Zn(OH)$_2$ protective layer, to further enhance anticorrosion properties [21].

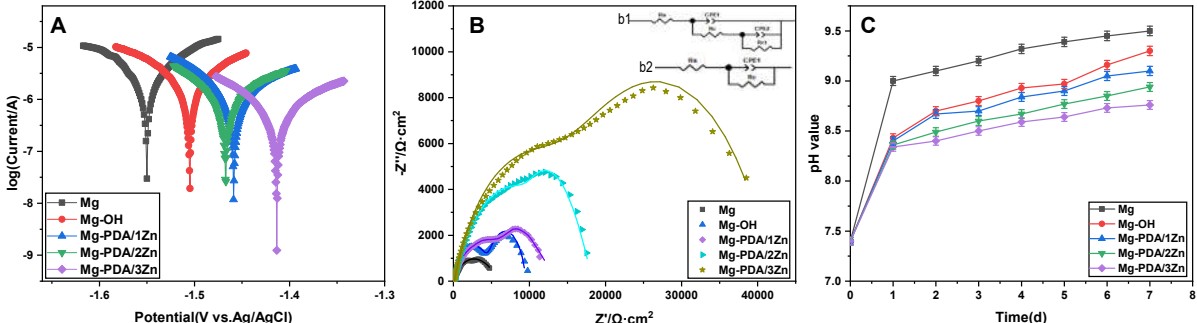

**Figure 3.** (**A**) Potentiodynamic scanning polarization curves and (**B**) Nyquist plots of the different specimens. For Nyquist plots, the scatter is the original values; the line is the calculative values; (b1) is equivalent circuits of Mg, Mg-OH, Mg-PDA/1Zn, and Mg-PDA/3Zn; and (b2) is equivalent circuits of Mg-PDA/2Zn. (**C**) pH changes in the different specimens immersed in SBF for 7 days.

According to the corrosion mechanism of the Mg alloy, plenty of OH$^-$ may be generated after corrosion, resulting in the pH value increasing, which can reflect the degradation behavior of the magnesium alloy. Figure 3C shows the pH values of the different Mg alloys immersed in SBF solution for 7 days. Obviously, the pH values of all specimens increased during this period. The pH value of the unmodified magnesium alloy sample was always the highest. In contrast, the trend of rising pH in the modified samples was relatively slow, indicating that this modification (alkali–heat treatment and PDA/Zn coating) can effectively reduce the release of OH$^-$ to a certain extent. Compared with PDA/Zn$^{2+}$-coated

modified magnesium alloys, the pH of Mg-OH was relatively high, because the Cl$^-$ in SBF can turn Mg(OH)$_2$ into soluble MgCl$_2$ after prolonged immersion, leading to the relatively fast degradation of the magnesium alloys. Notably, the pH value of Mg-PDA/3Zn was always the smallest during the 7 d immersion, demonstrating that it had the best anticorrosion properties and the slowest degradation rate, which can be attributed to the fact that the formation of Zn(OH)$_2$, and the robust PDA coating [22] were helpful for inhibiting the generation of OH$^-$, and reducing the pH value changes.

**Table 2.** Corrosion potential, corrosion current density, and annual corrosion depth of the different samples.

| Samples | $E_{corr}$/(V) | $i_{corr}$/(A·cm$^{-2}$) | d/(mm/y) |
|---|---|---|---|
| Mg | −1.55 | $1.42 \times 10^{-5}$ | $2.8 \times 10^{-1}$ |
| Mg-OH | −1.50 | $1.47 \times 10^{-6}$ | $7.95 \times 10^{-2}$ |
| Mg-PDA/1Zn | −1.45 | $9.23 \times 10^{-7}$ | $4.63 \times 10^{-2}$ |
| Mg-PDA/2Zn | −1.47 | $7.79 \times 10^{-7}$ | $4.30 \times 10^{-2}$ |
| Mg-PDA/3Zn | −1.41 | $4.41 \times 10^{-7}$ | $2.32 \times 10^{-2}$ |

**Table 3.** EIS fitting values of the different magnesium alloys.

| Samples | $R_s$/($\Omega$·cm$^2$) | $CPE_1$/($\mu$F·cm$^{-2}$) | $R_{ct}$/($\Omega$·cm$^2$) | $CPE_2$/($\mu$F·cm$^{-2}$) | $R_c$/($\Omega$·cm$^2$) | $R_p$/($\Omega$·cm$^2$) |
|---|---|---|---|---|---|---|
| Mg | 70.9 | $5.47 \times 10^{-5}$ | 1608 | $2.17 \times 10^{-4}$ | 4292 | 5900 |
| Mg-OH | 89.3 | $4.90 \times 10^{-5}$ | 2568 | $3.14 \times 10^{-4}$ | 4153 | 6721 |
| Mg-PDA/1Zn | 55.3 | $1.61 \times 10^{-5}$ | 7362 | $1.06 \times 10^{-4}$ | 8799 | 16,161 |
| Mg-PDA/2Zn | 83.6 | $2.36 \times 10^{-5}$ | 18,095 | - | - | 18,095 |
| Mg-PDA/3Zn | 91.5 | $5.94 \times 10^{-7}$ | 20,374 | $3.14 \times 10^{-5}$ | 37,048 | 57,422 |

$R_s$ is solution resistance; $CPE_1$, $CPE_2$ are constant phase elements; $R_{ct}$ is coating resistance; $R_c$ is charge transfer resistance.

The corrosion morphologies of the different magnesium alloys soaked in SBF for 1 d, 3 d, 7 d, and 14 d, respectively, are exhibited in Figure 4, and the surface elemental compositions after 14 d immersion determined by EDS are displayed in Table 4. After 1 d, cracks occurred on the Mg surface because of its poor anticorrosion properties; in contrast, no obvious cracks occurred on the other samples, indicating that the anticorrosion properties of the modified samples were improved to varying degrees. After 3 d, corrosion products could be observed on the Mg-OH surface, which could be due to the Mg(OH)$_2$ coating having been damaged to form the secondary corrosion products. For the PDA/Zn$^{2+}$-coated modified samples, the corrosion morphologies after 3 d immersion did not change obviously, compared to immersion of 1 d. It can be concluded that the zinc with positive potential was less likely to lose electrons in aqueous solution than Mg, which could provide better thermodynamic stability for the PDA/Zn$^{2+}$ coatings. The zinc-loaded polydopamine coating could inhibit the corrosive chloride ions from reaching the magnesium substrate, and thus enhanced anticorrosion [23]. After 7 d, the corrosion of Mg and Mg-OH samples were further intensified; microcracks could be found on the surfaces of Mg-PDA/1Zn and Mg-PDA/2Zn. It was considered that the existence of microdefects might provide the way in which the corrosive medium could contact the surface to cause corrosion. After 14 d, significant corrosion occurred on all surfaces, with the emergence of white particles and accumulation of corrosion products on the Mg and Mg-OH surfaces. Conversely, the Mg-PDA/3Zn surface exhibited minimal white particles, thereby signifying its superior corrosion resistance. The EDS results shown in Table 4 indicate that P and Ca occurred on all surfaces. The severity of corrosion observed on the Mg surface was significant, presenting as a discernible reduction in the Mg content, and an increase in the contents of C, O, and P, suggesting that the main component of the corrosion products could be a mixture of carbides and phosphides. The appearance of phosphorus on the Mg-OH surface indicated that the Mg(OH)$_2$ layer was damaged, potentially resulting in the creation of the Ca-P

compound. A small amount of Ca was found on the surfaces of the PDA/$Zn^{2+}$ coatings, which was due to the slight dissolution of the zinc-loaded polydopamine coating on the local area of the sample surface. The combination of $Zn^{2+}$ and $HPO_4^{2-}$ in the physiological environment could form Zn-based phosphates and oxides, which could reduce the chance of the magnesium substrate being directly exposed to aggressive solutions, and provide better corrosion protection for the magnesium substrate [22]. As displayed in Figure 4, as the concentration of zinc ions increased, the white particles on the surface decreased, and the Mg-PDA/3Zn surface was intact, with only a small amount of corrosion products. Taking all results into consideration, it can be concluded that Mg-PDA/3Zn had the highest coating integrity, and the best corrosion resistance.

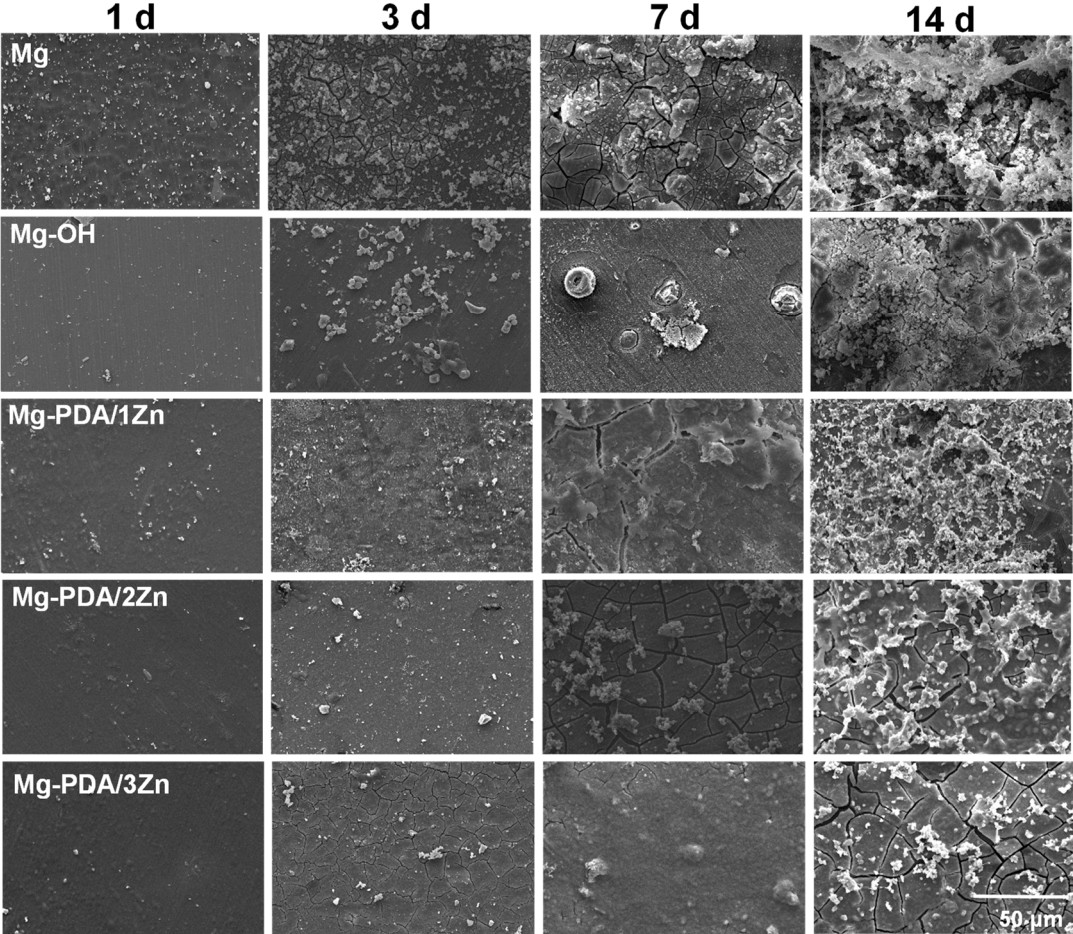

**Figure 4.** Representative SEM pictures of the different specimens incubated in SBF for 1 d, 3 d, 7 d, and 14 d, respectively.

**Table 4.** Surface element composition of the different specimens after 14 d immersion.

| Samples | Mg | C | O | P | Ca |
|---------|-----|------|------|------|------|
| Mg | 2.1 | 10.1 | 56.1 | 5.2 | 26.5 |
| Mg-OH | 1.5 | 26.8 | 39.2 | 15.0 | 17.5 |
| Mg-PDA/Zn | 3.3 | 27.7 | 53.6 | 6.8 | 8.6 |
| Mg-PDA/2Zn | 1.5 | 46.8 | 42.8 | 3.7 | 5.2 |
| Mg-PDA/3Zn | 1.1 | 37.1 | 44.5 | 7.6 | 9.7 |

*3.3. Surface Wettability and Protein Adsorption Behaviors*

Surface wettability has a key influence on protein adsorption and interactions between biomaterials and cells/blood. Protein adsorption is normally the first event for

blood-contact biomaterials after implantation, which participates in mediating biological responses, such as blood compatibility and cell behaviors. Albumin and fibrinogen, as two important proteins in human plasma, play an important role in hemocompatibility. Generally, albumin adsorption can inhibit platelet adhesion and activation, leading to enhanced blood compatibility. However, fibrinogen adsorption and denaturation can cause platelet activation and aggregation, to form a thrombus [24]. Figure 5 shows the results for the water contact angle and protein adsorption of the different samples. The water contact angle of the blank Mg alloy was 49°, and it decreased to 25° after alkali–heat treatment, resulting in better wettability. It was considered that plenty of the hydrophilic -OH could be introduced onto the surface using the NaOH treatment, leading to improvements in the surface wettability and antifouling performance [25]. After the fabrication of the PDA/$Zn^{2+}$ coatings, the water contact angle decreased significantly; this is because polydopamine with a large number of oxygen-containing groups can combine with hydrogen atoms in water to form a dense hydration layer, to achieve a better hydrophilic surface [26]. At the same time, zinc ions can also increase the hydrophilicity [27]. The hydrophilic surface exhibits a high capacity for water molecule absorption, which can minimize non-specific protein adsorption, and reduce platelet adhesion, which can contribute to improvement in the surface anticoagulant performance [28].

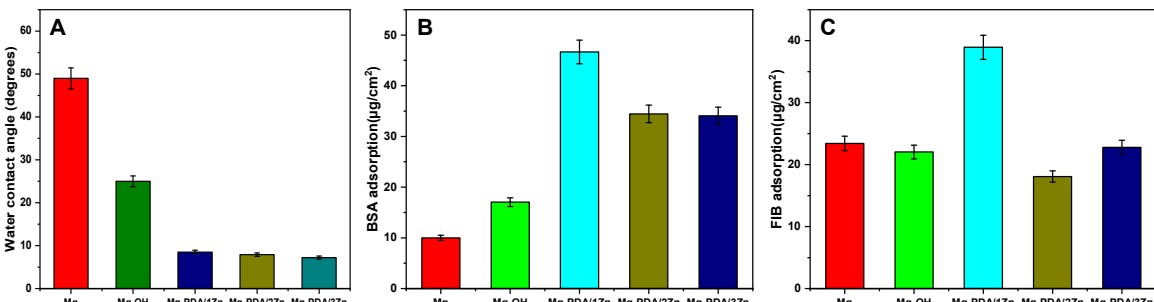

**Figure 5.** Water contact angle (**A**), albumin (**B**), and fibrinogen (**C**) adsorption amount of the different sample surfaces.

Compared with the unmodified magnesium alloys, the Mg-OH demonstrated excellent hydrophilicity, which contributed to BSA adsorption on the surface. In addition, fibrinogen has a strong affinity for hydrophobic surfaces [29], so the amount of fibrinogen adsorption on the Mg-OH surface decreased. Studies have indicated that polydopamine coating can significantly enhance protein adsorption, and $Zn^{2+}$ can also selectively promote albumin adsorption and improve hemocompatibility [27]. Compared with the Mg-OH, the fibrinogen adsorption on the PDA/$Zn^{2+}$ coating was higher. On the one hand, the hydroxyl groups on the surface could promote positively-charged fibrinogen adsorption [30]; on the other hand, the fibrinogen adsorption was more influenced by the surface roughness than the albumin adsorption was [31]. The surface became rougher after $Zn^{2+}$ loading, leading to the increase in fibrinogen adsorption. The imine and quinine groups of the PDA coating would rapidly adsorb proteins [32], which may be the reason for the higher adsorption capacity of Mg-PDA/1Zn than of Mg-PDA/2Zn and Mg-PDA/3Zn. Although the PDA coating could simultaneously enhance albumin and fibrinogen adsorption, the increase in albumin adsorption was more obvious, which could inhibit the fibrinogen adsorption to some degree, thereby reducing the chance of platelet adhesion and thrombus formation [33].

### 3.4. Blood Compatibility

Platelet adhesion behaviors on the surfaces of biomaterials can lead to coagulation, and even thrombosis and atherosclerosis [34]. Therefore, platelet adhesion assays are usually utilized to examine the hemocompatibility of biomaterials. Figure 6A shows the adhesion and quantity of the platelets adhered to the different surfaces. Obviously, many

platelets adhered to the Mg surface, accompanied by platelet aggregation and activation, suggesting that the hemocompatibility of the pristine Mg alloy was limited. This was related to its higher water contact angle, poor corrosion resistance, and a large amount of fibrinogen adsorption. The quantity of platelets on the Mg-OH surface decreased because the improved wettability was conducive to inhibiting the adsorption of plasma proteins and platelets; however, the quantity of platelets on the Mg-OH surface was still about $17,320/mm^2$, suggesting that its anticoagulation properties needed to be further enhanced. For PDA/$Zn^{2+}$-coated modified magnesium alloy, due to its excellent hydrophilicity and characteristic of selectively promoting albumin adsorption, the quantity of attached platelets was further reduced, which contributed to a reduction in thrombus formation. The quantity of attached platelets on the Mg-PDA/3Zn surface was higher than that on the Mg-PDA/2Zn surface, which may be due to the high level of FIB adsorption on the Mg-PDA/3Zn surface (Figure 5C).

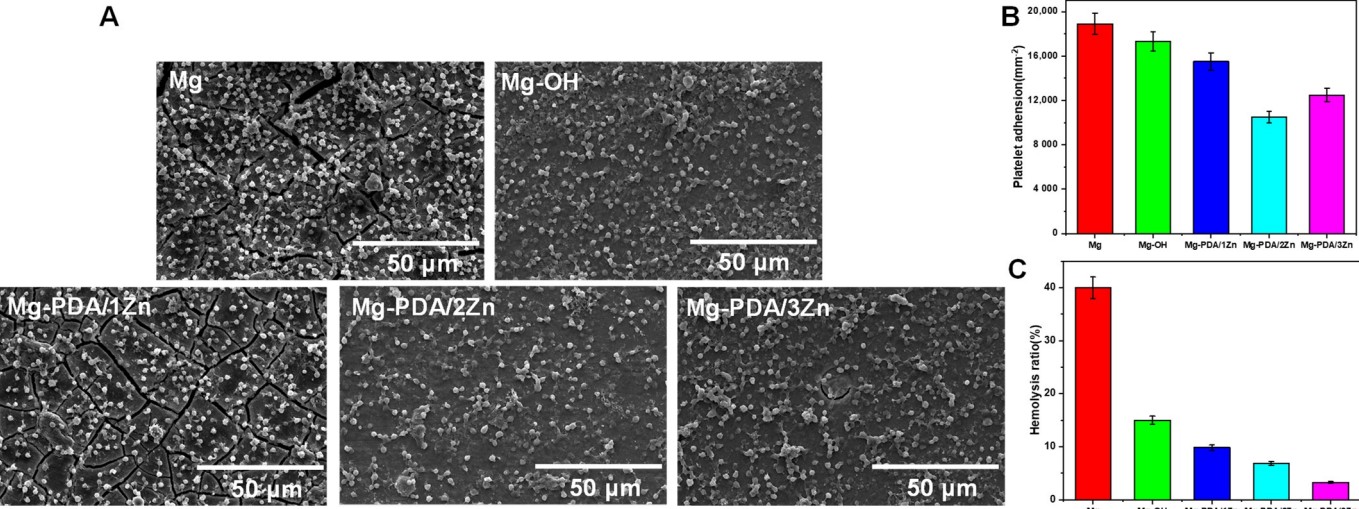

**Figure 6. (A)** Representative SEM pictures, and **(B)** quantity of platelets attached to the different specimens. **(C)** Hemolysis rates of the different specimens.

Figure 6C shows the hemolysis rates of the different Mg alloys. According to the standard of ISO 10,994.4: 2002, a hemolysis rate below 5% can be accepted in blood-contacting biomaterials. The released $OH^-$ caused by the corrosion of the pristine Mg alloy increased the pH value of the RBC suspension, thereby promoting the combination of hemoglobin and the cell membrane, which contributed to red blood cell rupture, with a hemolysis rate of 40%. This degree of hemolysis could potentially lead to severe complications. For Mg-OH, the hemolysis rate displayed a considerable reduction, due to the enhanced corrosion resistance; however, the value still exceeded 5%. After the preparation of the PDA/$Zn^{2+}$ coatings, the further improvement in corrosion resistance could significantly reduce the release of $OH^-$ and $Mg^{2+}$; at the same time, zinc ions have a certain protective effect on red blood cells, which can reduce the fragility of red blood cells, and maintain enzyme activity [35]. As a result, the hemolysis rate of $Zn^{2+}$-loaded polydopamine coating was significantly reduced. The hemolysis rate of Mg-PDA/3Zn was only 4.9%, which met the requirement for blood-contacting biomaterials (<5%).

### 3.5. Endothelial Cell Growth Behaviors

The interaction of cells with their surrounding environment plays a major role in the performance of the implant. Cell adhesion represents the first cellular event when cells make contact with the implant; it is a key factor that determines the performance of an implant in a clinical application [36]. Figures 7 and 8A are the fluorescent images and the CCK-8 values, respectively, of attached cells on the different surfaces. It can be clearly seen that whether cultured for 6 h or 24 h, the number of cell adhesions, and the CCK-8 value on

the Mg surface were the lowest, implying that the cells on the pristine Mg surface failed to differentiate and proliferate. On the one hand, the limited anticorrosion of the unmodified Mg alloys could lead to the formation of $H_2$ bubbles, and a rising pH environment, which are not helpful for cell adhesion and growth; on the other hand, the pristine Mg alloy surface had no bioactivities, and lacked cell adhesion sites to induce EC adhesion and spreading. The number of cells and the CCK-8 value for the Mg-OH increased slightly, and there was a greater increase at 24 h than at 6 h, which can be attributed to the enhanced anticorrosion properties and better hydrophilicity of the Mg-OH. After the preparation of the PDA/$Zn^{2+}$ coatings on the surface, whether the culture time was 6 h or 24 h, the quantity of adhered cells and the CCK-8 value on the surfaces had increased compared with Mg and Mg-OH, indicating that cytocompatibility had been significantly improved. With the increase in $Zn^{2+}$ concentration, the number of cell adhesions, and the CCK-8 value on the PDA/$Zn^{2+}$ modified surfaces also increased. However, due to the lower content of zinc ions on Mg-PDA/1Zn, the CCK-8 values of 6 h and 24 h were always slightly lower than those of Mg-PDA/2Zn and Mg-PDA/3Zn, implying that the zinc ions released from the PDA/$Zn^{2+}$ coatings could promote endothelial cell proliferation. There was no significant difference for the CCK-8 values between Mg-PDA/2Zn and Mg-PDA/3Zn, indicating that high concentrations of $Zn^{2+}$ did not consistently promote endothelial cell proliferation [37]. Studies have shown that the polydopamine coating can introduce a large quantity of functional groups (amine and catechol) to combine with many biomolecules, thereby enhancing cell adhesion and proliferation [38]. Ku et al. [39] confirmed that the polydopamine layer can minimize the deformation of serum proteins by regulating the surface energy to promote cell adhesion and proliferation. In addition, zinc ions are important trace elements to maintain cell redox balance. When the concentration is lower than 80 μM, it can enhance EC growth [40]. It is worth noting that the concentration of $Zn^{2+}$ in this study (3.4~10.4 μM, 1~3 mg/mL $ZnSO_4 \cdot 7H_2O$) was in this range. Therefore, better growth behavior of endothelial cells on the PDA/3Zn coating could be achieved.

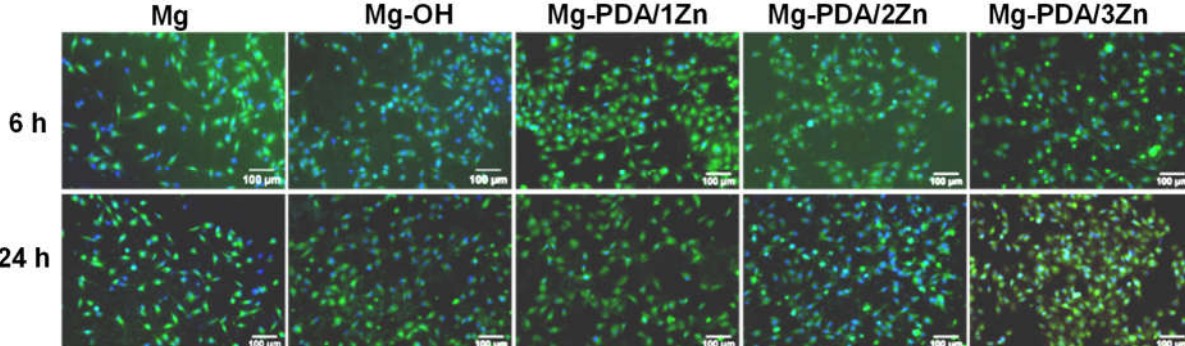

**Figure 7.** The fluorescence pictures of cells attached to the different surfaces for 6 h and 24 h, respectively.

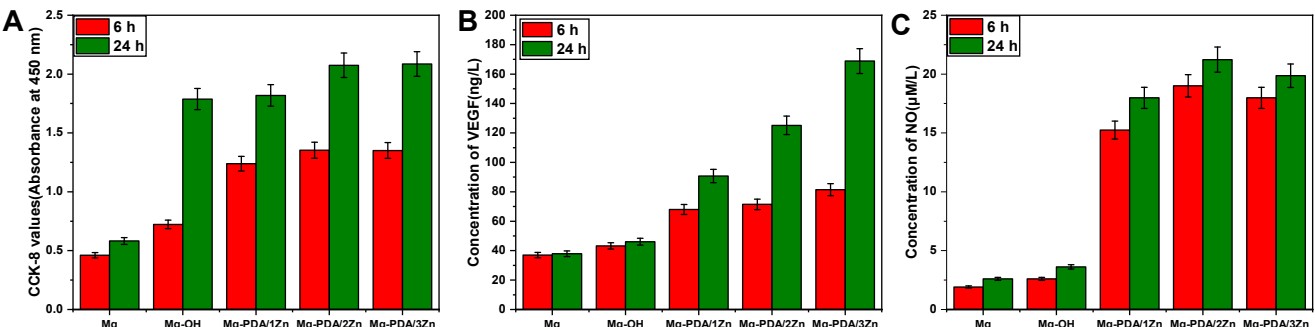

**Figure 8.** (**A**) CCK-8 values, (**B**) VEGF, and (**C**) NO expression of ECs cultured on the different specimens for 6 h and 24 h, respectively.

VEGF and NO are two key regulatory signaling molecules in the cardiovascular system, which can improve EC growth and proliferation, inhibit the migration and proliferation of smooth muscle cells, and prevent the adhesion and activation of platelets, reduce the formation of intravascular restenosis and intimal hyperplasia [41]. Endothelial cells in the human body can secrete VEGF and NO. The former can stimulate new angiogenesis, increase vascular permeability, and further enhance EC migration, proliferation, and differentiation; the latter can inhibit platelet aggregation, adhesion, and activation, regulate leukocyte adhesion and migration, and inhibit thrombosis and neointimal hyperplasia [42]. Figure 8B,C are the VEGF and NO expression of the ECs cultured on the different surfaces for 6 h and 24 h, respectively. It can be seen that the expression concentrations of VEGF (37 ng/L) and NO (2.5 μM/L) on the Mg surface were the lowest among all samples, indicating its poor endothelial growth performance. After alkali–heat treatment, the hydrophilicity and corrosion resistance were improved, which was beneficial to the adsorption and activity maintenance of proteins in plasma, leading to better endothelial cell adhesion. However, the VEGF and NO concentrations of Mg-OH were only increased to 46 ng/L and 3.9 μM/L, respectively. After constructing the PDA/$Zn^{2+}$ coatings, the expression concentrations of VEGF and NO were significantly increased. With the prolongation of the culture time, the ECs can express more NO and VEGF, thereby promoting EC growth. On the one hand, a small amount of fibrinogen adsorbed on the PDA/$Zn^{2+}$ coatings helped ECs to express more VEGF, and promoted endothelial cell proliferation [43]; on the other hand, the incorporation of $Zn^{2+}$ improved the activity of NO enzymes inside cells, which contributed to the upregulated NO expression [44]. With the increase in zinc ion concentration, the increase in VEGF and NO expression had a significant effect on maintaining cell function and promoting endothelial cell differentiation, which was of great significance in improving the cytocompatibility of the Mg alloys. It is worth noting that from the perspective of NO expression (Figure 8C), Mg-PDA/1Zn was slightly lower than Mg-PDA/2Zn and Mg-PDA/3Zn, and there was no significant difference between Mg-PDA/2Zn and Mg-PDA/3Zn, suggesting that the continuous release of $Zn^{2+}$ in the coating, to further improve EC differentiation, was difficult. Therefore, taking the results of cell behaviors into consideration, Mg-PDA/3Zn was the best at promoting EC adhesion, proliferation, and function expression.

## 4. Conclusions

In this paper, PDA/$Zn^{2+}$ coatings containing different $Zn^{2+}$ concentrations were successfully prepared on alkali–heat-treated magnesium alloys through the self-polymerization of dopamine, and its chelation to zinc ions. The properties and functions of the materials before and after the modification were studied, and the conclusions are as follows:

(1)  The $Zn^{2+}$-loaded PDA coatings displayed superior characteristics with respect to corrosion. With the increase in $Zn^{2+}$ in the coating, the anticorrosion properties of the Mg alloy were enhanced, and the Mg-PDA/3Zn had the best anticorrosion properties.

(2)  The PDA/$Zn^{2+}$ coatings not only displayed excellent hydrophilic properties and corrosion resistance, but could also release zinc ions, which could provide a favorable surface for selectively promoting albumin adsorption and significantly improving hemocompatibility.

(3)  Due to the sustained release of $Zn^{2+}$ from the PDA/$Zn^{2+}$ coatings, the endothelial cell adhesion and proliferation, as well as the VEGF and NO expression, were enhanced.

(4)  Due to the PDA coating's capability to further react with bioactive molecules with different functions, its use is expected in further enhancing the corrosion resistance and biocompatibility of the Mg alloys, by immobilizing bioactive molecules on the surface.

**Author Contributions:** Conceptualization, L.M. and C.P.; methodology, L.M.; software, L.M., Y.J. and L.W.; validation, Q.Z.; formal analysis, L.M. and J.C.; investigation, L.M. and Q.H.; resources, Q.Z. and C.P.; data curation, X.L.; writing—original draft preparation, L.M. and C.P.; writing—review and editing, J.C., Q.Z. and C.P.; visualization, Q.H. and J.C.; supervision, J.C. and C.P.; funding acquisition, C.P. All authors have read and agreed to the published version of the manuscript.

**Funding:** This work was financially supported by the National Natural Science Foundation of China (31870952) and the Natural Science Foundation of the Jiangsu Higher Education Institutions of China (No. 21KJB430013).

**Data Availability Statement:** Not applicable.

**Conflicts of Interest:** The authors declare no conflict of interest.

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
