# Peer review of "Fabrication of Zn2+-Loaded Polydopamine Coatings on Magnesium Alloy Surfaces to Enhance Corrosion Resistance and Biocompatibility"

_coatings, doi:10.3390/coatings13061079_

Round 1
Reviewer 1 Report
This paper discusses a new approach used to improve the use of magnesium alloys in biodegradable stents for heart conditions. Currently, these alloys degrade too quickly and aren't biocompatible enough for use in the body. Pan and coworkers tackled these issues by applying a coating made from zinc and polydopamine, onto the alloy. This coating improves the alloy's interaction with blood, prevents it from degrading too rapidly, and promotes better integration with body cells.
Overall the paper is well-structured, systematically explores the different manufacturing conditions and includes typical techniques for the characterisation of the surfaces (ATR-FTIR, XPS, STM, potentiometric analysis) etc.
Minor correction suggestions:
- The introduction can benefit from a scheme depicting the alloy manufacturing steps.
- References seem not to be all formatted in the same/similar way
- Please shorten the paper title and make it more direct (e.g. Zn-loaded Polydopamine coatings on magnesium-allow surfaces)
- Abstract: Sentence 1 / Magnesium alloys (plural) .. its good mechanical properties (singular).
- Abstract: Sentence 1 / phrases such as "more and more" re repetitive, can substitute with "increasing attention"
Author Response
This paper discusses a new approach used to improve the use of magnesium alloys in biodegradable stents for heart conditions. Currently, these alloys degrade too quickly and aren't biocompatible enough for use in the body. Pan and coworkers tackled these issues by applying a coating made from zinc and polydopamine, onto the alloy. This coating improves the alloy's interaction with blood, prevents it from degrading too rapidly, and promotes better integration with body cells.
Overall the paper is well-structured, systematically explores the different manufacturing conditions and includes typical techniques for the characterisation of the surfaces (ATR-FTIR, XPS, STM, potentiometric analysis) etc.
Minor correction suggestions:
- The introduction can benefit from a scheme depicting the alloy manufacturing steps.
Re:We gratefully appreciate for your valuable comment. We have added some content based on your suggestion in the introduction. “In the past several years, many kinds of magnesium alloy with excellent mechanical properties have been developed. For example, Zhang et al developed a Mg alloy with ultrahigh ductility (near 50%) by combining Er micro-alloying (0.3 at% Er) with appropriate grain refinement (∼8 µm) [4]. Zhao et al. [5] utilized the super divine wave welding technology to achieve high-strength connections between two heterogeneous light alloys (magnesium alloy and copper). However, most of the magnesium alloys are chemically active and their biodegradation in the complicated physiological environment of the human body easily leads to the premature loss of mechanical support after the implantation.”
Reviewer 2 Report
In this paper, the Zn2+-loaded polydopamine (PDA/Zn2+) coating was constructed on the alkali-heat-treated magnesium alloy through the chelating effect of PDA with metal ions. This was done to to improve the corrosion resistance and biocompatibility. It is shown that the PDA/Zn2+ coatings with different Zn2+ contents could not only endow the magnesium alloys with the excellent hydrophilicity and selectively promotion of the albumin adsorption, but also significantly improve the corrosion resistance in the simulated physiological environment. The corrosion resistance of the modified magnesium alloy was enhanced with the increase of Zn2+ content in the coating. The paper could be considered for publication in the journal of Coatings after the following major revisions:
1-Move these sentences to the introduction section:
“Magnesium alloys are attracting more and more attention in the field of biodegradable cardiovascular stent due to its good mechanical properties and biodegradability. However, the rapid degradation after the implantation and the limited biocompatibility restrict its clinical applications. Zinc is the second trace element in the human body and has a significant impact on the biological behavior of cells and tissues.”
2-Define in the abstract what parameters were investigated, before briefly mentioning the results of such tests.
3-Check the English of the whole paper. Too many connection words are used, e.g. meanwhile, moreover etc.
4-Introduciton should be strengthened. To modify this section the following documents can be consulted:
-(2023). Developing a Mg alloy with ultrahigh room temperature ductility via grain boundary segregation and activation of non-basal slips. International Journal of Plasticity, 162, 103548. doi: -https://doi.org/10.1016/j.ijplas.2023.103548
-(2023). Ultrasonic welding of AZ31B magnesium alloy and pure copper: microstructure, mechanical properties and finite element analysis. Journal of Materials Research and Technology, 23, 1273-1284. doi: https://doi.org/10.1016/j.jmrt.2023.01.095
5-Materials and methods should be shortened. Some of the experimental explanation are known and need not be repeated.
6-what is the y axis in figure 1a,b?
7-in table 1, reporting only one digit is enough. Round the values to one digit after the dot.
8-consult the following references in the discussion section:
-(2022). Phase transformations in an ultralight BCC Mg alloy during anisothermal ageing. Acta Materialia, 239, 118248. doi: https://doi.org/10.1016/j.actamat.2022.118248
-(2023). Antifouling Coatings Fabricated by Laser Cladding. Coatings, 13(2). doi: 10.3390/coatings13020397
9-what do Mg and MgOH figure in figure 2 show? Figure 2 could be shortened or even deleted.
10-Conclusions should be shortened and provide in bullet points.
11-better describe the rationale of the work at the end of the introduction. The last paragraph of the introduction is badly started.
12-too many figures are provided in figure 4. Some of them could be deleted.
Check the English of the whole paper. Too many connection words are used, e.g. meanwhile, moreover etc.
Author Response
In this paper, the Zn2+-loaded polydopamine (PDA/Zn2+) coating was constructed on the alkali-heat-treated magnesium alloy through the chelating effect of PDA with metal ions. This was done to improve the corrosion resistance and biocompatibility. It is shown that the PDA/Zn2+ coatings with different Zn2+ contents could not only endow the magnesium alloys with the excellent hydrophilicity and selectively promotion of the albumin adsorption, but also significantly improve the corrosion resistance in the simulated physiological environment. The corrosion resistance of the modified magnesium alloy was enhanced with the increase of Zn2+ content in the coating. The paper could be considered for publication in the journal of Coatings after the following major revisions:
1 Move these sentences to the introduction section:
“Magnesium alloys are attracting more and more attention in the field of biodegradable cardiovascular stent due to its good mechanical properties and biodegradability. However, the rapid degradation after the implantation and the limited biocompatibility restrict its clinical applications. Zinc is the second trace element in the human body and has a significant impact on the biological behavior of cells and tissues.”
Re: We sincerely thank your valuable feedback. The corresponding sentence in the Abstract has been removed.
2 Define in the abstract what parameters were investigated, before briefly mentioning the results of such tests.
Re: We think this is an excellent suggestion. We have revised the abstract according to the suggestion.
3 Check the English of the whole paper. Too many connection words are used, e.g. meanwhile, moreover etc.
Re: We feel great thanks for your professional review work on our article. According to your comment, we have made extensive corrections to our previous draft.
4 Introduction should be strengthened. To modify this section the following documents can be consulted:
-(2023). Developing a Mg alloy with ultrahigh room temperature ductility via grain boundary segregation and activation of non-basal slips. International Journal of Plasticity, 162, 103548. doi: -https://doi.org/10.1016/j.ijplas.2023.103548
-(2023). Ultrasonic welding of AZ31B magnesium alloy and pure copper: microstructure, mechanical properties and finite element analysis. Journal of Materials Research and Technology, 23, 1273-1284. doi: https://doi.org/10.1016/j.jmrt.2023.01.095
Re: We appreciate your valuable comments. We have made changes to the corresponding parts of the manuscript.
Some sentences have been revised and the phrase “In the past several years, many kinds of magnesium alloy with excellent mechanical properties have been developed. For example, Zhang et al developed a Mg alloy with ultrahigh ductility (near 50%) by combining Er micro-alloying (0.3 at% Er) with appropriate grain refinement (∼8 µm) [4]. Zhao et al. [5] utilized the super divine wave welding technology to achieve high-strength connections between two heterogeneous light alloys (magnesium alloy and copper).” has been added to strength the Introduction.
5 Materials and methods should be shortened. Some of the experimental explanation are known and need not be repeated.
Re: We gratefully appreciate for your valuable comment. We have refined the experiments and methods in the paper before submitting it. For the completeness of the paper, we suggest retaining the necessary information.
6-what is the y axis in figure 1a,b?
Re: The y-axis in Figures 1a and b represent transmittance and intensity, respectively. To facilitate comparison of multiple curve peaks, a Y offset stacked line graph is used to place multiple curves into a single layer, clearly displaying the data for each group, so that the Y-axis of different groups has a relative offset.
7-in table 1, reporting only one digit is enough. Round the values to one digit after the dot.
Re: We appreciate your valuable comments. We have revised the relevant issues that arise in this article.
8 consult the following references in the discussion section:
-(2022). Phase transformations in an ultralight BCC Mg alloy during anisothermal ageing. Acta Materialia, 239, 118248. doi: https://doi.org/10.1016/j.actamat.2022.118248
-(2023). Antifouling Coatings Fabricated by Laser Cladding. Coatings, 13(2). doi: 10.3390/coatings13020397
Re: We appreciate your valuable comments. The first literature reveals the complete sequence of phase transformation in alloys and provides morphological evidence for the mechanism of phase transformation, which is not consistent with the research content of this paper. It is difficult for us to consult the references for the discussion section.
Phrase “It was considered that a large number of the hydrophilic groups (-OH) on the surface produced by the alkali heat treatment can lead to the improvement of the surface wettability and antifouling performance [25].” has been added in the section “3.3. Surface Wettability and Protein Adsorption Behaviors”.
9 what do Mg and MgOH figure in figure 2 show? Figure 2 could be shortened or even deleted.
Re: The surface morphologies of unmodified magnesium alloy and alkali heat treated magnesium alloy are shown by Mg and Mg OH in Figure 2. Although the surface differences between the Mg and Mg-OH samples are not significant, we recommend retaining figure 2 for the sake of the logic and soundness of the paper.
10 Conclusions should be shortened and provide in bullet points.
Re: We appreciate your valuable comments. We have made the revisions in the conclusion section.
11 better describe the rationale of the work at the end of the introduction. The last paragraph of the introduction is badly started.
Re: We think this is an excellent suggestion. We have made changes to the corresponding parts of the manuscript.
12 too many figures are provided in figure 4. Some of them could be deleted.
Re: We sincerely thank you for careful reading. In order to provide readers with a clearer understanding of the corrosion behaviors of the different magnesium alloys soaked for 14 days, we believe it is necessary to provide the surface corrosion morphologies for 1d, 3d, 7d, and 14d.
Round 2
Reviewer 2 Report
The paper can be published in its revised format.